# Machine learning accurate exchange and correlation functionals of the electronic density

Sebastian Dick 🔬 [1,2✉] & Marivi Fernandez-Serra 🔬 [1,2✉]

Density functional theory (DFT) is the standard formalism to study the electronic structure of matter at the atomic scale. In Kohn–Sham DFT simulations, the balance between accuracy and computational cost depends on the choice of exchange and correlation functional, which only exists in approximate form. Here, we propose a framework to create density functionals using supervised machine learning, termed NeuralXC. These machine-learned functionals are designed to lift the accuracy of baseline functionals towards that provided by more accurate methods while maintaining their efficiency. We show that the functionals learn a meaningful representation of the physical information contained in the training data, making them transferable across systems. A NeuralXC functional optimized for water outperforms other methods characterizing bond breaking and excels when comparing against experimental results. This work demonstrates that NeuralXC is a first step towards the design of a universal, highly accurate functional valid for both molecules and solids.

[1] Physics and Astronomy Department, Stony Brook University, Stony Brook, NY 11794-3800, USA. [2] Institute for Advanced Computational Science, Stony Brook University, Stony Brook, Stony Brook, NY 11794-3800, USA. ✉email: sebastian.dick@stonybrook.edu; maria.fernandez-serra@stonybrook.edu

For many years, density functional theory (DFT) has served as the standard tool to study the electronic structure of materials and condensed systems. Striking an optimal balance between accuracy and computational cost[1], DFT makes a first-principles description of complex and large systems possible that is otherwise out of reach for more accurate ab initio approaches. To achieve this balance, DFT is mapped onto a mean-field single-electron description within the Kohn–Sham (KS)[2] approach. In KS–DFT, all the complexities of the many-body electron–electron interaction are reduced within a functional of the density. This functional consists of an exchange (X) and a correlation (C) part, the former capturing effects from Pauli exchange, and the latter approximating correlations of electrons within the many-body wave function.

There is a well-defined roadmap to creating more accurate XC functional formulations, the so-called Jacob's ladder of John Perdew[3,4], with each rung representing increasing levels of complexity and decreasing levels of approximation to the exact XC functional. The construction of functionals following this map allows for the incorporation of the added complexities in a controlled and physically motivated way, imposing the necessary constraints that these formulations should satisfy to correctly and universally describe the underlying physics.

A completely different approach to obtaining more accurate functionals is to replace the physically motivated path by a data-driven search. Functionals created following this approach are often referred to as semiempirical[5], and versions of these functionals implement approximations from all rungs of the aforementioned Jacob's ladder. In recent years, unprecedented computational capacity has made the calculation of physical properties of molecules and solids with ab initio fully correlated accuracy possible. Such developments have allowed researchers to take the semiempirical approach to the extreme, inaugurating an era of machine learning (ML) methods in density functional development. This path produced the recent $\omega$B97M-V[6], a range-separated hybrid meta-GGA with nonlocal correlation. It was designed using a combinatorial technique taking Becke's B97 family of semiempirical functionals[7], augmented with hybrid and nonlocal correlation components as primary ingredients. The fit was done using a database of accurate single-point calculations on a few thousand molecules. Similarly, using a simple mathematical formulation coined data projection on the parameter subspace, Fritz et al.[8] showed that it was possible to optimize a GGA functional with nonlocal correlations for liquid water. This functional was fitted to highly accurate data from coupled-cluster calculations that were also used to optimize the water force field MB-pol[9–11].

While these latter functionals can already be considered members of the ML family, other modern ML approaches make use of algorithms such as artificial neural networks (ANN), kernel ridge regression, and Gaussian process regression. Grifasi et al.[12] have shown that the electron density for small hydrocarbons can be directly predicted from structural information and Fabrizio et al.[13] have been able to extend this work to noncovalently bonded systems. Chandrasekaran et al.[14] were able to achieve the same goal for solid-state systems by introducing a grid-based structure to electron density mapping using an ANN. Both approaches show great promise to significantly speed up ab initio calculations as they completely circumvent solving the cubic-scaling self-consistent field (SCF) equations. Other works, including the one presented here, have attempted to parametrize an XC functional with ML, and we discuss related methods[15–17] in detail in Supplementary Note 1.

In this manuscript, we propose a pathway to construct fully machine-learned functionals that depend explicitly on the electronic density and implicitly on the atomic positions and are built on top of physically motivated functionals in a $\Delta$-learning type approach. These functionals are created for a specific dataset and hence are not universal. They follow the philosophy of other optimized density functionals[8], which opt to prioritize the system-dependent accuracy over their transferability. We will show that using our proposed method, it is possible to create specialized functionals that perform close to coupled-cluster level of accuracy when used in systems with sufficient similarity to the training data. Functionals exhibit promising transferability from gas to condensed phase and from small to larger molecules within the same type of chemical bonding. Moreover, far outside their training domain, these functionals are shown not to decrease the accuracy of their baseline method.

Our method is an evolution of our recent work[18], in which we developed machine-learned correcting functionals (MLCFs) to correct energies and forces by learning from the electron density. Building on it, in this manuscript, we show that it is possible to take the functional derivative of MLCFs thus creating semilocal ML KS density functionals that can be used in self-consistent calculations. We call this overall method NeuralXC. We show that these functionals encode meaningful chemical information that extends beyond the training set, hence making the functionals transferable. Despite not using the density as a target in the training process, we discuss how the resulting self-consistent densities compare to the exact (at the coupled cluster with singles, doubles and perturbative triples (CCSD(T)) level) densities. Except for some specific moments of the density distribution, we do not observe a major improvement. We discuss approaches to overcome this limitation, which will be further developed in future work.

## Results

**Density representation.** The charge density is represented following our earlier work[18] by projecting it onto a set of atom-centered basis functions. Throughout this work the inner cutoff radius was set to zero, resulting in radial basis functions defined as

$$\tilde{\zeta}_n(r) = \begin{cases} \frac{1}{N} r^2 (r_o - r)^{n+2} & \text{for } r < r_o \\ 0 & \text{else} \end{cases} \quad (1)$$

with an outer cutoff radius $r_o$ and a normalization factor $N$. The full basis is then given by $\psi_{nlm}(\mathbf{r}) = Y_{lm}(\theta, \phi)\zeta_n(r)$, where $Y_{lm}(\theta, \phi)$ are real spherical harmonics and $\zeta_n$ the orthogonalized radial basis functions (for details see ref. [18]). The basis set parameters chosen for every model used in this work are summarized in Supplementary Table 1.

The descriptors $c_{nlm}^I$ for atom $I$ of species $\alpha_I$ at position $\mathbf{R}_I$ are obtained by projecting the electron density $\rho$ onto the corresponding basis functions $\psi_{nlm}^{\alpha_I}$:

$$c_{nlm}^I \equiv c_{nlm}[\rho(\mathbf{r}), \mathbf{R}_I, \alpha_I] = \int_{\mathbf{r}} \rho(\mathbf{r})\psi_{nlm}^{\alpha_I}(\mathbf{r} - \mathbf{R}_I). \quad (2)$$

We found it beneficial for certain models to use the modified electron density $\delta\rho$ instead of $\rho$ in Eq. (2). This $\delta\rho$ is defined as the difference between the full electron density and atomic electron density $\rho_{atm}$ the latter being constructed by filling the basis functions with appropriate valence charges (see ref. [19] for details):

$$\delta\rho(\mathbf{r}) = \rho(\mathbf{r}) - \rho_{atm}(\mathbf{r}). \quad (3)$$

Using this neutral density has the advantage that it is generally smoother than $\rho$, as peaks around the ion cores cancel out. Moreover, $\delta\rho$ always integrates to zero, regardless of the atomic species involved, suggesting that models trained on it will show better transferability across chemical environments. We have used $\delta\rho$ in all models introduced below except for the one trained on water clusters. Here, cross-validation has determined $\rho$ to

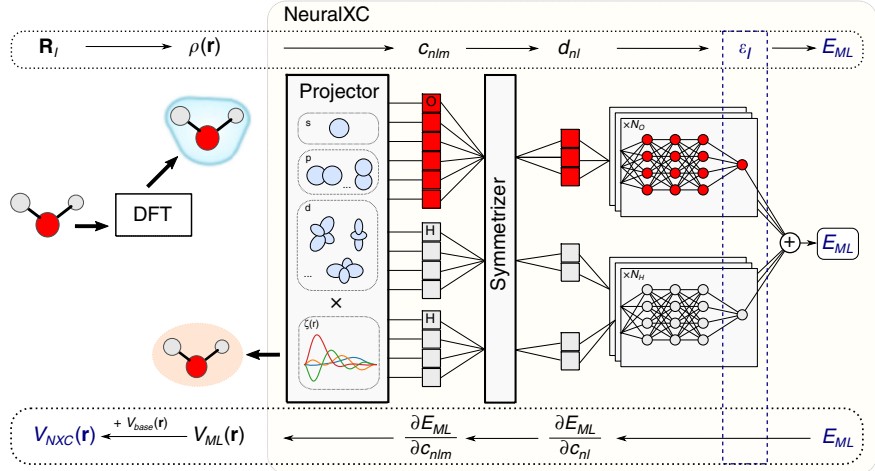

**Fig. 1 Implementation of NeuralXC.** Starting from the electron density in real space, obtained with a converged DFT calculation (using the baseline functional $E_{base}$), the projector maps this density to a set of descriptors $c_{nlm}$. The symmetrizer creates rotationally invariant versions of these descriptors $d_{nl}$, which, after preprocessing (not depicted here), are passed through a Behler–Parrinello type neural network architecture. By using the same network for descriptors of a given atomic species, we ensure permutation invariance. Once the energy $E_{ML}$ is obtained, its derivative can be backpropagated using the chain rule to obtain the machine-learned potential $V_{ML}$. $V_{ML}$ is added back to the baseline potential $V_{base} = \delta E_{base}/\delta n(\mathbf{r})$, to create the full $V_{NXC}(\mathbf{r})$, which can be used in subsequent self-consistent calculations.

produce lower generalization errors (see "Methods" section for details).

To avoid erroneous behavior during deployment, the model must respect all physical symmetries. These symmetries include permutations of atoms of the same species, rotations, and reflections. We opted to enforce these symmetries in two ways: permutational invariance is imposed by the architecture of our neural network as discussed below, whereas rotational invariance and invariance under reflection is encoded in the features themselves.

Starting from our original descriptors $c_{nlm}$, we can obtain a rotationally invariant version by applying the transformation

$$d_{nl} = \sum_{m=-l}^{l} c_{nlm}^2. \tag{4}$$

**Machine-learned functional**. As in previous work by the authors[18], the permutationally invariant Behler–Parrinello networks (BPN)[20] were used to parametrize the energy functional. The network maps the rotationally invariant descriptors $d_{nl}$ onto the energy, which is represented as a sum of atomic contributions to ensure permutation symmetry (Fig. 1). The energy functional can therefore be written as

$$E_{ML}[\rho(\mathbf{r})] = E_{ML}(\mathbf{d}[\rho(\mathbf{r})]) = \sum_I \epsilon_{\alpha_I}(\mathbf{d}[\rho(\mathbf{r}), \mathbf{R}_I, \alpha_I]), \tag{5}$$

where $\epsilon_\alpha$ are the outputs of the atomic networks, i.e. the last layer inside the BPN before the global summation. We have further used $\mathbf{d}_I$ as a short-hand notation for the collection of $d_{nl}$ over all allowed values for $n$ and $l$.

The functional is built on top of a physically motivated, non-ML baseline functional $E_{base}$, which in this work was chosen to be PBE[21]. Other choices for this baseline functional are possible but will lead to a different trade-off between accuracy and computational cost.

Once the energy functional has been fitted, the potential $V_{ML}$, which is required to perform self-consistent calculations, can be obtained through

$$V_{ML}[\rho(\mathbf{r})] = \frac{\delta E_{ML}[\rho]}{\delta \rho(\mathbf{r})}. \tag{6}$$

Here, $\frac{\delta}{\delta\rho(\mathbf{r})}$ indicates the functional derivative and should not be confused with the modified electron density in Eq. (3). Together with Eq. (2) this translates to

$$V_{ML}[\rho(\mathbf{r})] = \sum_\beta \frac{\partial E_{ML}}{\partial c_\beta} \frac{\delta c_\beta[\rho]}{\delta \rho(\mathbf{r})} = \sum_\beta \frac{\partial E_{ML}}{\partial c_\beta} \psi_\beta(\mathbf{r}). \tag{7}$$

Here, we have used $\beta$ as a composite index, summarizing the indices $n$, $l$ and $m$ as well as the atomic index $I$. Using Eq. (4), the partial derivatives can be computed as

$$\frac{\partial E_{ML}}{\partial c_\beta} \equiv \frac{\partial E_{ML}}{\partial c_{nlm}} = 2 \frac{\partial E_{ML}}{\partial d_{nl}} c_{nlm}. \tag{8}$$

The resulting potential is therefore a linear combination of the original basis functions, with coefficients depending on the derivatives of the machine-learned energy functional with respect to its input features. These derivatives are usually implemented in ML software packages and thus straightforward to obtain. The machine-learned potential and energy are both added back to their baseline counterparts

$$E_{NXC}[\rho] = E_{base}[\rho] + E_{ML}[\rho], \tag{9}$$

$$V_{NXC}[\rho] = V_{base}[\rho] + V_{ML}[\rho]. \tag{10}$$

The combined functionals (NXC for NeuralXC) can in principle be used in any DFT code.

It is clear that the energy functional has an implicit dependency on both, the atomic species $\alpha_I$ and the nuclear coordinates $\mathbf{R}_I$, setting it apart from traditional semilocal functionals and some other ML approaches (see above). While the former dependency can be lifted by simply using the same basis set and atomic neural network for every atom regardless of species, the latter is inherent to our method and cannot be circumvented. Previous work on machine-learned kinetic energy functionals seems to indicate that encoding information about the atomic positions in the features can be beneficial[22].

At this point it should be highlighted that we do not create a single functional, but a collection of functionals. Each functional within this collection is trained on and therefore closely linked to a specific dataset. These datasets were chosen in order to test and illustrate certain properties of our proposed method. We named

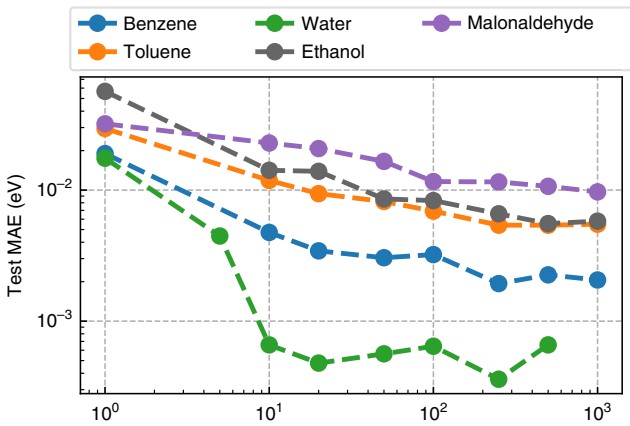

**Fig. 2 Mean average error.** Error in energy prediction on sGDML[25] test set with respect to training set size.

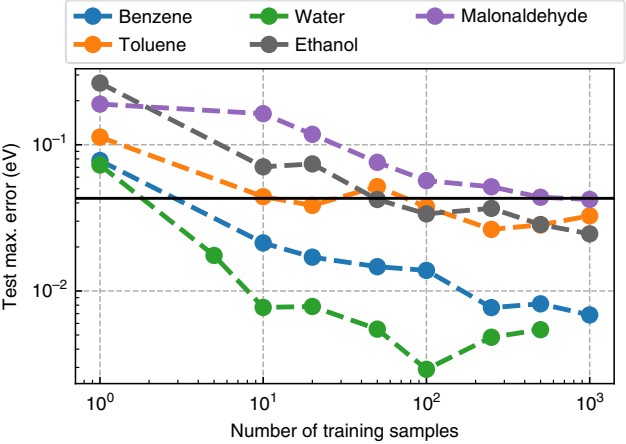

**Fig. 3 Maximum absolute error.** Error in energy prediction on sGDML[25] test set.

the three datasets used in this work after the methods they originate from: sGDML[23], MOB-ML[24] and MB-Pol[9–11]. These sets contain total energies for a variety of structures calculated at the coupled cluster with singles doubles and perturbative triples (CCSD(T)) level. For further details, we refer the reader to Supplementary Note 2.

**Data efficiency**. Frequently, training data is scarce or, as in our case, expensive to obtain. Due to the unfavorable scaling of correlated quantum chemistry methods, the creation of highly accurate datasets for medium to large-sized molecules remains challenging to this day. We would, therefore, like to design an ML method that utilizes information contained in the available training data to its full extent.

In order to test the data efficiency of NeuralXC, we trained an ML functional for every molecule contained in the sGDML dataset[23] while varying the amount of training data.

Figures 2 and 3 show how the generalization error changes as the size of the training set is increased. For each training set size, a new model was trained using the iterative approach described in the "Methods" section, and self-consistent calculations were run on the entire test set. We used two different metrics for the evaluation: the mean absolute error (MAE) and the maximum absolute error. It can be seen that the MAE starts to saturate at values of 0.01 eV or below at roughly 100 training samples. Some improvement in the maximum error can be observed as the training set size is increased further. For malonaldehyde, at least 500 samples are required to reach a max. error below chemical accuracy (1 kcal/mol or 0.043 eV), all other molecules pass that threshold at 100 samples or fewer.

**Transferability**. Beyond being data-efficient, a useful ML model generalizes well to unseen data. It is traditionally assumed that both training and test set are independent identically distributed samples of the same underlying distribution. There is no reason to believe that a model should extrapolate beyond the population on which it was trained.

In an apparent contrast to this, we would like to create a machine-learned functional that, after being exposed to a small sample of molecules, generalizes to more complex and larger systems. However, even though molecules might differ significantly in their structural variables from those contained in the training set, locally, their charge distributions and, therefore, the input to the network can still be similar as long as the underlying chemistry does not change too much.

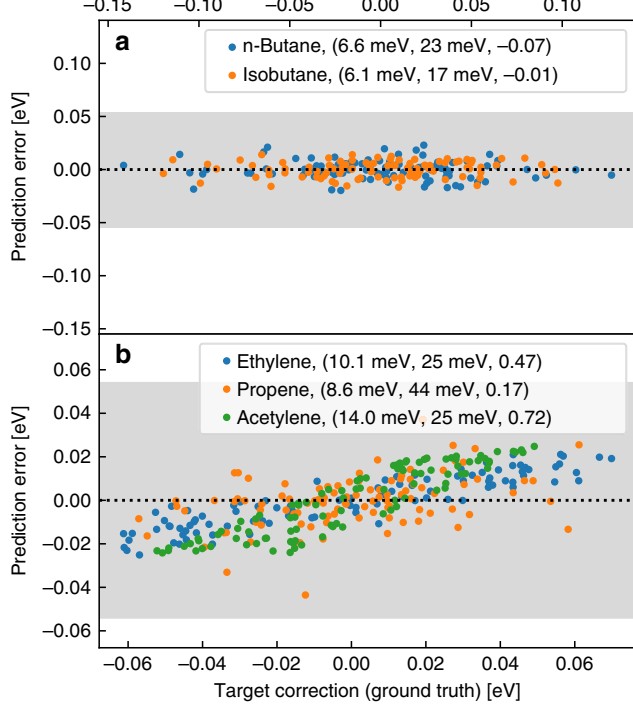

**Fig. 4 Residuals for transferability task on small alkanes. a, b** Values on the x-axis, the target corrections, correspond to the errors in total energy of PBE compared to the reference method CCSD(T). Prediction errors on the y-axis are defined as the errors of NeuralXC optimized on ethane and propane with respect to CCSD(T). Shaded area corresponds to an error of ±2mH, the threshold chosen by Cheng et al.[26]. Values in parentheses correspond to (mean absolute error, maximum absolute error, $R^2$ for the residuals).

To test the transferability of our functional, we start by comparing our method to that of Cheng et al.[26] using the MOB-ML dataset[24]. After being trained on 50 ethane and 20 propane geometries, the model's capability of correctly reproducing relative energies for 100 n-butane and isobutane geometries is assessed. Figure 4a shows that these energies are predicted well beyond chemical accuracy with MAEs of 6.6 and 6.1 meV, respectively, and that in fact, we are more accurate than Cheng et al.'s[26] state of the art method which achieves MAEs of 8.7 and 8.8 meV. Even after the training set size was decreased to 10 ethane and 5 propane structures, our model's accuracy remains comparable to that of Cheng et al.'s[26], as can be seen in Table 1.

**Table 1 Transferability task on small alkanes.**

| Method | Training set composition | Ethane | | Propane | | Butane | | Isobutane | |
|---|---|---|---|---|---|---|---|---|---|
| | | MAE | Max. | MAE | Max. | MAE | Max. | MAE | Max. |
| PBE | – | 15 | 70 | 12 | 47 | 9.3 | 29 | 9.0 | 30 |
| SCAN | – | 12 | 52 | 8.7 | 37 | 6.5 | 26 | 6.8 | 22 |
| $\omega$B97M-V | – | 8.5 | 41 | 6.3 | 26 | 4.8 | 18 | 4.5 | 18 |
| MOB-ML[27] | 100, 100, 50 | – | – | – | – | 4.0 | 15 | 6.3 | 22 |
| MOB-ML(mod.)[26] | 20, 50, 0 | – | – | – | – | 2.2 | 9.5 | 2.2 | 9.5 |
| NeuralXC | 20, 50, 0 | 2.0 | 9 | 1.8 | 7.7 | 1.7 | 5.8 | 1.5 | 4.3 |
| NeuralXC | 5, 10, 0 | 2.4 | 14 | 2.3 | 11 | 2.2 | 7.8 | 2.3 | 6.8 |

Mean absolute error (MAE) and maximum absolute error (Max.) per carbon atom. The second column describes how many samples of propane, ethane, and methane were contained in the training set. Energy errors are given in meV.

**Table 2 Generalization errors of NXC-W01.**

| Method | 1-body | | | 2-body | | | 3-body | | |
|---|---|---|---|---|---|---|---|---|---|
| | RMSE | MAE | Max. | RMSE | MAE | Max. | RMSE | MAE | Max. |
| PBE | 61 | 48 | 174 | 35 | 19 | 270 | 11 | 5.8 | 75 |
| SCAN | 9.2 | 7.7 | 22 | 44 | 24 | 297 | 13 | 7.7 | 49 |
| $\omega$B97M-V | 7.4 | 5.0 | 40 | 16 | 11 | 65 | 11 | 6.9 | 60 |
| NXC-W01 | 1.8 | 1.4 | 9.3 | 11 | 7.5 | 47 | 8.0 | 4.6 | 41 |

The errors in total energy are split up into their many-body contributions. For monomers the 1-body errors are reported, for dimers the 2-body errors and for trimers the 3-body errors. All values are given in meV.

Both specialized NeuralXC functionals as well as MOB-ML (mod.) outperform SCAN and $\omega$B97M-V (results obtained with PySCF and a cc-pVDZ basis), two state of the art functionals, in accuracy on the test data.

We would further like to assess how well our model generalizes to other hybridizations of the carbon atom. Figure 4b shows the prediction errors of the model used on an augmented test set containing systems with double and triple bonds. While we see a decline in performance for these systems, the model still improves upon PBE. In particular, errors in total energy are within 1.6 mHartree or 44 meV of the reference values. The linear correlation between prediction error and target value indicated by their large $R^2$ coefficients suggests the existence of systematic errors. These errors are most likely due to the model's failure to treat physical effects deriving from the $sp$ and $sp^2$ hybridizations of the carbon atom and could be compensated by including relevant structures in the training set. Indeed, we have found that by merely adding three ethylene structures to the training set, the $R^2$ coefficients for ethylene, propene, and acetylene decrease to 0.11, 0.01, and 0.30, respectively.

We have also tested how well our method generalizes to elements other than those contained in the training set. Ideally, we would like to create a general functional that can be used across a wide variety of elements. To do so, it is necessary to remove any information about the atomic species in the model input. While we predict that an extensive and carefully curated training will be necessary to achieve high accuracy across systems, we have shown in Supplementary Note 5 and Supplementary Table 2 that a species-independent NeuralXC can be trained on a set of O- and C-containing molecules and exhibit some improvement for molecules with S and Si. In particular, the average error in bond lengths for a set of small molecules decreased by ~42%.

**Condensed systems and molecular dynamics**. The previous test has focused on evaluating the transferability within single molecule gas-phase systems. A different transferability measure should evaluate the capacity of a functional trained on small clusters to describe condensed phase systems. We chose to test this by running Born–Oppenheimer molecular dynamics simulations of liquid water—a challenging system for standard DFT methods[8]—using the NeuralXC functional optimized on the MB-pol dataset[9–11].

The machine-learned functional was built as an additive correction to the PBE XC functional and consisted of a sum of two models. The first model was trained to jointly reproduce the total energies of monomers and dimers. The second model was then built on top of the first to correct three-body energies in trimers. We coin this new NeuralXC functional NXC-W01.

Table 2 shows the NXC-W01 generalization error compared to its baseline method on a test set consisting of 200 monomers, 500 dimers, and 250 trimers, obtained in the same way as the training set. Rather than comparing total energies, we show errors for one-, two-, and three-body energies as defined in ref. [9] as otherwise large contributions from one-body energies would always dominate the comparison. Moreover, it has been shown that the failure of common density functionals to reproduce the structure of liquid water can largely be accredited to the incorrect treatment of low-order many-body energies. Conversely, a functional that reproduces these energies with high confidence is expected to give an accurate description of liquid water[32]. We have further included results for the functionals SCAN and $\omega$B97M-V. These results were obtained using PySCF employing the cc-pVQZ basis set. Table 2 shows that NXC-W01 is superior to the other functionals tested.

In addition, we have tested the functional on the s66 dataset[33] to assess its transferability to heterogeneous systems. The results shown in Supplementary Table 3 and discussed in Supplementary Note 6 indicate that the functional improves the overall treatment of hydrogen bonds lowering the average error in bonding distance from 0.039(6) Å for PBE to 0.021(3) Å for NXC-W01.

Using our ML model as a potential instead of merely adding an energy correction as proposed in earlier work by the authors[18]

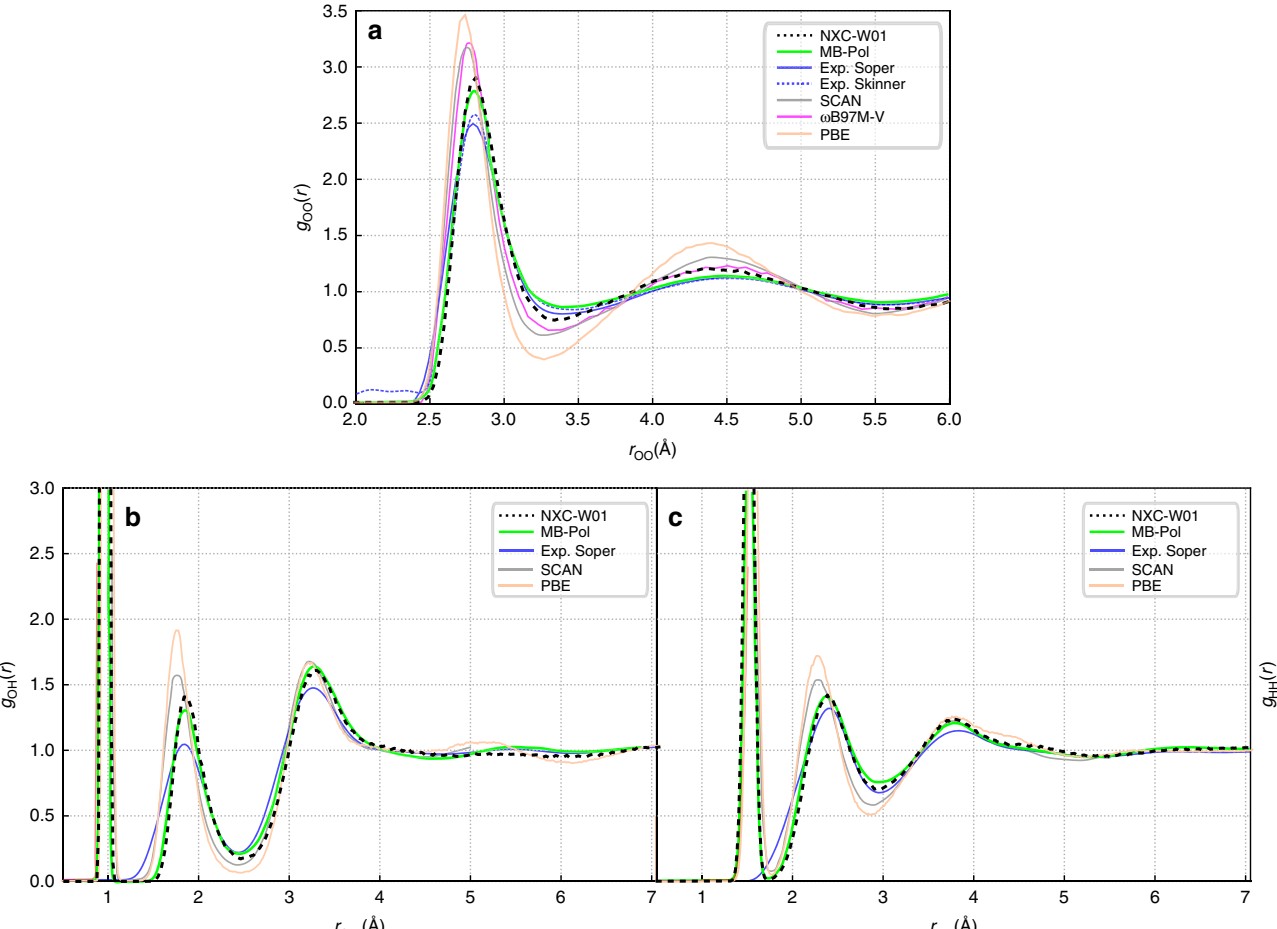

**Fig. 5 Radial distribution functions (RDFs). a–c** The RDFs obtained from Born–Oppenheimer molecular dynamics simulations of 96 water molecules in a periodic box at experimental density and 300 K using PBE and NXC-W01 as functionals are compared to experimental results by Skinner et al.[28] and Soper[29] as well as MB-Pol results taken from ref. [11] and Born–Oppenheimer MD simulations using SCAN by Wiktor et al.[30] and ωB97M-V by Yao et al.[31].

and in related work[15] has the advantage that electron densities are self-consistent with respect to the underlying functional. Self-consistency makes the Hellmann Feynman theorem[34] applicable, allowing us to obtain accurate, energy-preserving forces that can be used to study dynamical and statistical properties of a system.

It is commonly accepted that the accurate description of liquid water necessitates the use of hybrid functionals and the explicit treatment of dispersion forces and nuclear quantum effects (NQEs)[35]. The latter is often achieved through path integral molecular dynamics[36], the cost of which still prohibits its use in ab initio simulations of realistically sized systems. Testing our optimized functional on liquid water, we, therefore, bear in mind that an exact agreement with experimental results could only be achieved if NQEs were to be explicitly included.

Born–Oppenheimer molecular dynamics simulations were run for 96 water molecules in a periodic box at experimental density and 300 K using stochastic velocity rescaling as implemented by the i-PI code[37]. We obtained an initial configuration from a thermalized molecular dynamics simulation of the same system run with MB-pol. This configuration was then used together with random initial velocities as starting point for 20 ps MD runs with time step 0.5 fs, using both PBE and NXC-W01 as functionals. We discarded the first 5 ps and used the remaining 15 ps for our analysis.

As MB-Pol has been shown to provide excellent agreement with experimental results[11] in PIMD studies, the quality of our model can be assessed by comparing to MB-Pol classical molecular dynamics simulations at 300 K. We further include

results from various other works obtained with functionals that are considered superior to PBE, namely the *meta*-GGA functional SCAN and the range-separated hybrid functional with nonlocal interactions ωB97M-V. The results for SCAN were taken from work by Wiktor et al.[30] who conduct 15 ps long simulations with a time step of 0.48 fs, performed in the canonical NVT-ensemble at 300 K using the CP2K code and a periodic box containing 64 water molecules. For ωB97M-V, results by Yao et al.[31] were included, the computational details being the same as in the case of SCAN except for a total simulation time of 30 ps and a time step of 1.5 fs. For radial distribution functions (RDFs) other than oxygen–oxygen, only results by Wiktor et al.[30] were available.

Figure 5 shows excellent agreement between the RDFs obtained with NXC-W01 and MB-pol. While both SCAN and ωB97M-V show improvement with respect to PBE, both functionals lead to an overstructured liquid. This is in accordance with the insights presented in Table 2, as we would expect NXC-W01 to outperform the other functionals based on its accuracy regarding many-body energies. While deviations with respect to x-ray diffraction experiments[28] and joint refinement of neutron and x-ray data[29] can be observed, these can be largely accredited to the lack of explicit treatment of NQEs. Quantities that are more robust to these effects such as the shape of the first trough as well as the radial positions of extrema in all RDFs are well reproduced.

We have also validated that NXC-W01 is capable of accurately describing bond breaking situations in water, which it was not

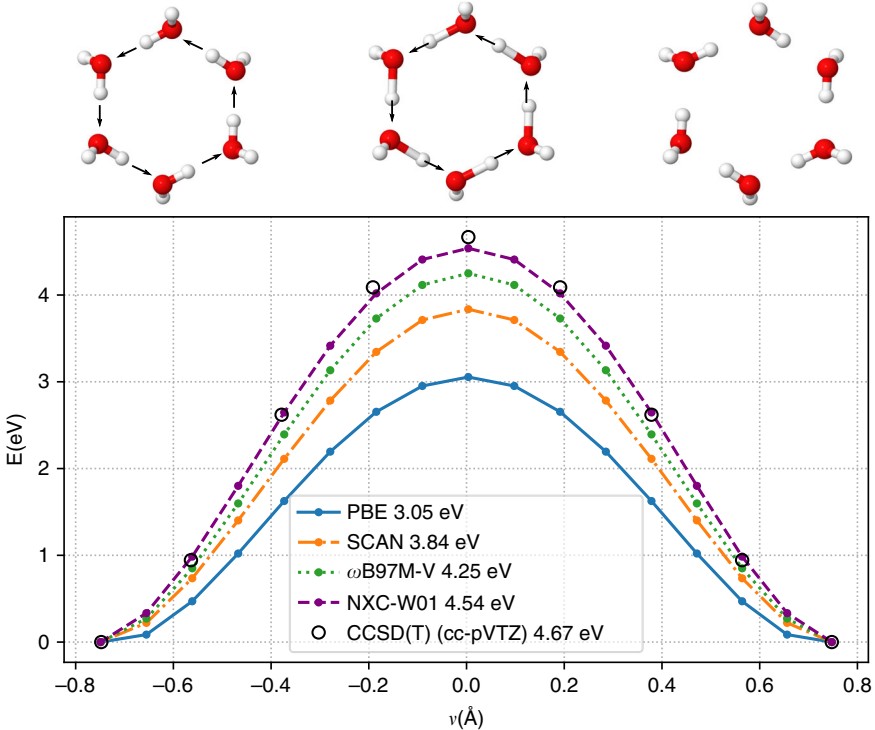

**Fig. 6 Coordinated proton transfer in a water-hexamer ring.** The reaction coordinate is defined as $\nu = d(O-H) - d(O'-H)$, where $O, O'$ are the two oxygen atoms involved in the H bond and H is the transferred proton. Energy values shown in inset correspond to barrier heights.

explicitly trained for. Figure 6 shows the coordinated proton transfer reaction in a water-hexamer ring. A simultaneous six-proton transfer path along the H bond direction between the six molecules in the ring is discretized and the energy at each configuration is plotted as a function of the proton transfer coordinate $\nu$. None of our training data involved dissociated configurations, however NXC-W01 outperforms all other XC functionals and closely reproduces CCSD(T) results. These type of dissociative configurations are explored by ring polymer beads in path integral molecular dynamics simulations of liquid water[38], and cannot be accounted for with nondissociative force fields.

**Electronic densities.** When trying to evaluate the quality of electron densities produced by NXC-W01, we are faced with the problem of comparing densities that were obtained with different methods and approximations.

In particular, coupled cluster densities were calculated with PySCF[39], an all electron code that utilizes Gaussian basis sets and nonperiodic boundary conditions. In contrast, NeuralXC is implemented within SIESTA[19], a periodic, pseudopotential DFT code that uses numerical atomic orbital basis sets. This limits the meaningfulness of a density comparison based on real-space grids.

We hence choose to compare the moments of the density distribution. These (dipole and quadrupole moments) have a direct physical interpretation and have been used before to evaluate the quality of a given DFT density[40]; moreover, they are accessible by experiment. Table 3 shows the dipole and quadrupole moments, together with the spread of the valence electron density distribution for a water molecule in the experimental equilibrium geometry. Results evidence that NXC-W01 improves the moments of the density distribution. Particularly, the dipole moment error of PBE is reduced from 2 to 0.2% with NXC-W01.

Figure 7 shows the valence charge density changes with respect to the fixed baseline model for a water molecule in its experimental geometry. Additional density comparisons for other molecules and functionals are provided in Supplementary Fig. 6.

**Table 3 Moments of the electronic density of water.**

|  | Exp. | PBE | NXC-W01 | CCSD(T) |
|---|---|---|---|---|
| Dipole (D) | 1.855 | 1.814 | 1.851 | 1.856 |
| Quadrupole $Q_T$ (D × Å) | 2.565 | 2.488 | 2.494 | 2.505 |
| $\langle r^2 \rangle$ (D × Å) | – | −26.83 | −26.45 | −26.51 |

Calculations were done for a molecule in its experimental equilibrium geometry. Coupled-cluster results were calculated in an aug-cc-pVTZ basis, a doubly polarized quadruple zeta basis was used for PBE and NXC-W01. The quadrupole moment $Q_T$ is defined as half the difference between the largest and smallest eigenvalue of the traceless quadrupole tensor $Q_T = 1/2(q_{max} - q_{min})$. It is invariant to rotations and uniquely defines the entire quadrupole tensor for a water molecule in its equilibrium geometry[41]. The spread of the valence electron density is defined as $\langle r^2 \rangle = \int_r r^2 \rho_{val}$.

The plotted density cuts show that there is qualitative agreement between the two method mostly along the OH bond where both methods localize more charge than PBE. Closer to the oxygen core, the change in density induced by NXC-W01 exhibits a nodal shape that is missing in the exact counterpart.

To understand the source of these deviations, it is instructive to revisit Eq. (7): In DFT, the ground state density is uniquely determined by the potential $V$. In the case of NeuralXC, $V_{ML}$ is closely related to the derivatives of the atomic neural networks with respect to their input features. In regions of feature space where data is abundant, fitting the model to reference energies will give a valid treatment of these derivatives (assuming that the neural network is sufficiently smooth, which can be achieved with regularization techniques). However in data-sparse regions, these derivatives will become less reliable.

Returning to the example of water and Fig. 7, it becomes clear why NXC-W01 achieves a satisfying treatment of the OH bonds, as density variations within that area are well represented in the training data. As the density close to the oxygen core is less susceptible to molecular deformations, especially when using pseudopotentials, NXC-W01 has less data to draw upon in this region.

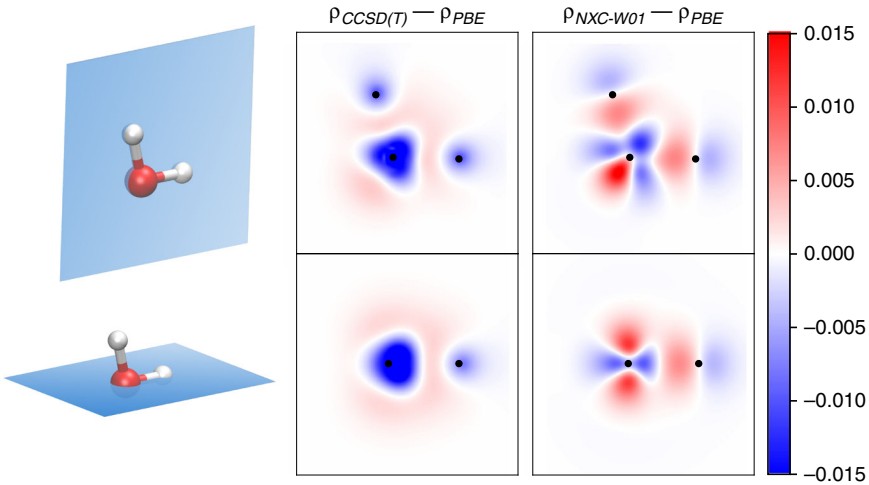

**Fig. 7 Electron density of water.** Comparison of the difference in electron density between CCSD(T) and PBE and NXC-W01 and PBE for a water molecule in its experimental equilibrium geometry. Two dimensional cuts either correspond to high-symmetry planes or planes containing a significant number of atoms and are indicated by blue surfaces in the molecule depictions adjacent to the density plots. Black dots inside the density plots indicate the positions of in-plane atoms. Atoms are color-coded with red corresponding to oxygen and white to hydrogen. Color scale is in units of $e \times Bohr^{-3}$.

The question remains whether a machine-learned XC functional of the form of NeuralXC can be brought closer to an exact functional. One way to improve in this regard is to incorporate information about the exact potential during the training process. This potential can in principle be calculated starting from the exact density obtained from a fully correlated many-body wave function. While we aim to explore this in more detail in future work, we provide a proof of concept example in Supplementary Note 7, showing how the density error of PBE can be decreased by up to two orders of magnitude for a set of $H_2$ molecules .

## Discussion

We have developed a supervised ML method termed NeuralXC that lifts the accuracy of KS density functional calculations at a GGA level towards that of coupled-cluster theory calculations. We have shown that using NeuralXC, it is possible to create specialized functionals that are highly accurate when used in systems sufficiently similar to their training data, while not degrading the overall accuracy of their baseline method (and in some cases improving it) when used far outside their training domain.

Throughout this work we have tried to illustrate several key aspects that, we believe, contribute to the success of an ML method.

Given the limited availability of highly accurate reference data, it is crucial that the proposed method is data-efficient. We have shown that desired accuracies for a variety of systems can be reached with small to moderately sized training sets.

Another cornerstone of a successful ML model is its transferability, facilitating model creation itself. As NeuralXC functionals generalize across chemical environments, the need to create a new reference dataset and retrain a new model for each system of interest decreases. We have shown this in the case of alkanes, where a model trained on ethane and propane was still valid for n-butane and isobutane structures showing an MAE in total energy prediction of 6.6 and 6.1 meV, respectively. While showing promising results, these experiments also laid bare the shortcomings of our method, as the functional only proved limited capability of treating carbon hybridizations other than the one it was trained on.

In comparison to other models presented in this work, which were used as case studies to highlight certain strengths and weaknesses of our method, NXC-W01 stands out as a versatile

functional with promising future applications. Beyond reproducing pair-correlation functions close to experimental results, it is capable of treating bond breaking, opening the path to studying proton transfer processes in liquid water at a highly accurate level. Further, we have shown that beyond water, the model is capable of correcting the hydrogen bond length for a variety of systems contained in the s66 dataset. For systems where NXC-W01 does not provide an improvement it was shown that it does not significantly degrade the accuracy of its baseline functional, PBE. This suggests that NXC-W01 can be used in scenarios where the correct treatment of water–water interactions is crucial and PBE is known to have sufficient accuracy for the remaining interactions. For example, it is a suitable model to treat hybrid systems like aqueous interfaces, or solutions, where the water description is highly sensitive to the quality of the functional.

All these insights will guide further development of our method, the ultimate goal being the design of a universal functional that is equally valid and highly accurate for a wide variety of systems both from the realm of molecules and solids. The success of this endeavor will depend crucially on the availability of diverse and accurate training data.

Furthermore, while this has not always been done in the past[42], density functionals should be judged by their ability to reproduce both energetic benchmarks as well as reference electron densities. This work has put an emphasis on energetic properties, but we have shown that by correcting the total baseline energy NeuralXC also induces density changes that bring the density closer to the exact density. However, as these changes are relatively small, future research will need to address how reference potentials can be directly incorporated in the training procedure to enable a more guided approach towards functionals that are accurate regarding both energy and density.

NeuralXC opens up a new path to developing exchange–correlation functionals for KS–DFT calculations. As our method only introduces a linearly scaling overhead to the underlying baseline functional (see Supplementary Note 4), it is especially attractive for simulations of large systems for which explicitly correlated wave function methods are still too expensive. Beyond creating accurate functionals for KS–DFT calculations, we see possible applications in orbital-free DFT, where NeuralXC could be used to develop kinetic energy functionals.

Finally, the trade-off between accuracy and cost that our method entails needs to be carefully assessed. This trade-off depends both on the baseline functional and the basis sets used (as well as other variables). While being somewhat ad hoc and less physically motivated, we have previously shown[18] that ML density functionals can also be used to correct for basis set errors. Building a NeuralXC functional on top of a cheap baseline such as the local density approximation[43], together with a minimal basis set, could make our method a competitive alternative to tight-binding DFT.

## Methods

**Training**. The models were trained on self-consistent densities produced with the baseline functional (PBE[21]). Given a dataset containing triplets of the baseline total energies $E_{base}^{(i)}$, reference total energies $E_{ref}^{(i)}$, and baseline densities $\rho^{(i)}$, the loss function is defined as

$$\mathcal{L} = \sum_i^N \left( E_{ref}^{(i)} - E_{NXC}[\rho^{(i)}] \right)^2, \tag{11}$$

$$= \sum_i^N \left( (E_{ref}^{(i)} - E_{base}^{(i)}) - E_{ML}[\rho^{(i)}] \right)^2, \tag{12}$$

where the parameters inside the machine-learned functional $E_{ML}$ are to be optimized to minimize $\mathcal{L}$.

Before passing the symmetrized descriptors $d_{nl}$ through the neural network, three additional preprocessing steps were employed. First, a variance filter was used, disregarding all features whose variance across the training set was below a threshold value equal to $10^{-10}$, effectively de-noising the dataset. Second, all features are scaled so that their values are normally distributed across the training set with zero mean and variance one, a step common in ML to ensure fast convergence of the optimization algorithm used to train the neural network. As a final step, the features were projected onto their principal components[44], only keeping enough components so that an explained (normalized) variance of $\gamma$ was achieved, with values of $\gamma$ ranging from 0.95 to 1. If $\gamma$ is smaller than one, this step has a regularizing effect decreasing the risk of overfitting.

All models were implemented in Tensorflow[45] and trained using the Adam[46] optimizer with training rate $\alpha = 0.001$ and decay rates $\beta_1 = 0.9$ and $\beta_2 = 0.999$ and the sigmoid function was chosen as activation. Hyperparameters such as $\gamma$, the learning rate, l2-regularization were determined through k-fold cross-validation. This involves splitting the training data into k random folds, i.e. equally sized parts, and picking the hyperparameters that produce the smallest average generalization error on a single fold if trained on the remaining ones. Once these hyperparameters are determined, the model is trained one final time on the entire training set. We used $k = 5$ for training sets with less than 100 data points and $k = 3$ for all others. Supplementary Note 3 discusses how a model architecture could in principle be optimized for maximum transferability.

The number of nodes per hidden layer was also treated as a hyperparameter and optimized through cross-validation. The final depth (i.e., the number of hidden layers) for each network was not explicitly chosen as it was determined by the convergence of the iterative training procedure described below. A summary of the resulting network architectures is given in Supplementary Table 1.

By altering the XC functional, the self-consistent electron densities change as well. This fact causes the actual accuracy of the ML functional, defined as the accuracy of the energies and forces obtained by self-consistent calculations with the modified functional, to be lower than the accuracy obtained during the fitting procedure. To remedy this, we employed what we call iterative training: The electron densities and corrected energies obtained with the first iteration of the ML functional $E_{ML}^{(1)}$ are used to train a new iteration which is then in turn used to calculate new densities. This procedure is continued until the accuracy of the obtained functional remains unchanged across two subsequent iterations. The neural network used in iteration $n + 1$ is obtained by freezing the hidden layers of iteration $n$ and adding a new hidden layer to the network that is then optimized on the $n^{th}$ iteration of the training densities. Typical numbers of iterations (and final number of hidden layers) ranged from two to five. This technique is reminiscent of a procedure commonly known as greedy layer-wise training in the deep learning community[47], although with a different goal set. A more detailed discussion of the training algorithm can be found in the Supplementary Methods.

**DFT calculations**. The baseline calculations for all of the datasets above were conducted with SIESTA[19] using the PBE[21] exchange–correlation functional with norm-conserving pseudopotentials, a real-space grid cutoff of 400 Ry and a cubic unit cell with lattice constant 30 Å unless otherwise indicated. A doubly polarized quadruple zeta basis set was used for the water clusters and the s66x8 dataset calculations. All other structures were computed with a polarized double zeta basis. Molecular dynamics simulations were conducted using an optimized polarized double zeta basis[48] and a real-space grid cutoff of 450 Ry.

**ML basis sets**. The ML basis sets were hand-picked using a combination of physical intuition (to set reasonable lower and upper bounds for the parameters) and cross-validation. The basis set used for MOB-ML was optimized for transferability. This was achieved by training models on methane and ethane and determining which basis parameters produce the best extrapolation (lowest RMSE) to propane. The basis sets used are listed in Supplementary Table 1.

## Data availability

Data in the form of molecule geometries along with their associated reference energies, as well as input files and scripts needed to reproduce the results presented in this manuscript, are bundled with our initial release of NeuralXC and available in zenodo under the indentifier https://doi.org/10.5281/zenodo.3761613[49]. Additional data related to this paper may be requested from the authors.

## Code availability

The implementation of NeuralXC as well as examples on how to train and deploy NeuralXC functionals are available in zenodo with the identifier https://doi.org/10.5281/zenodo.3761613[49].

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

## Acknowledgements

This work was supported by the U.S. Department of Energy, Office of Science, Basic Energy Sciences, under Awards DE-SC0001137 and DE-SC0019394, as part of the CCS and CTC Programs. S.D. was partially supported by a fellowship from The Molecular Sciences Software Institute under NSF grant ACI-1547580. We would like to thank Stony Brook Research Computing and Cyberinfrastructure, and the Institute for Advanced Computational Science at Stony Brook University for access to the high-performance SeaWulf computing system, which was made possible by a $1.4M National Science Foundation grant (#1531492). S.D. wants to express his thanks to Samuel Ellis for his valuable advice regarding the implementation of NeuralXC. Finally, NeuralXC uses an implementation of the gradient of spherical harmonics implemented in SIESTA[19].

## Author contributions

S.D. conceived the idea and performed the experiments. M.F.-S. and S.D. designed the experiments and analyzed the data. M.F.-S. supervised the research. Both authors contributed equally to writing the manuscript.

## Competing Interests

The authors declare no competing interests.
