## [Peer Review File · Nature Communications]

Reviewers' comments:

Reviewer #1 (Remarks to the Author):

Dick and Fernandez-Serra discuss the development of density functionals based on supervised machine learning by using CCSD(T) energies as reference. Their aim is to provide a highly-accurate density (and energy) at the speed of a calculation performed with a GGA functional. In this manuscript, they present an interesting method that uses atomic orbitals as feature space of the input electron density. A neural network is then trained that can predict a new, more accurate density. However, the method has already been introduced in a previous publication, while the two manuscripts share many similarities, including test examples and figures. In addition, the authors mention in every section (including the title) the word "accuracy", but their results do not fully reflect that statement. For those reasons, I cannot recommend this article for publication in Nature Communications.

Technical comments:

- The title is misleading. Reference data from CCSD(T) obtained with a triple-zeta quality basis set should not be called as "accurate". There might be more accurate than GGA functionals, but in order to obtain the expected accuracy of post-HF methods and especially coupled-cluster methods, a complete basis set is needed (basis set extrapolation, explicitly correlated methods).
- I do not understand why the authors are trying to map a GGA density generated with planewaves and pseudopotentials to data from cluster CCSD(T) calculations. Since all calculations have been performed for molecular species, why they do not use a cluster DFT code?
- The computational details for the CCSD(T) calculations are pretty messy. No quantum chemical code is mentioned for the generation of reference data for the sGCML and MB-pol. In addition, using different reference data from different codes and approximations (eg. density fitting), makes the analysis of the results imbalanced.
- Efforts to satisfy invariance by data augmentation is inefficient.
- In the section "data-efficiency", I don't understand where the efficiency is if you need a dataset of at least 500 samples to reach 1 kcal/mol (malonaldehyde).
- Abstract and introduction mention that density functionals have been developed, so I was expecting one unified functional as the outcome of the ML methodology. On the contrary, the manuscript describes the development of case-specific functionals.
- The example of cats and dogs should be removed.
- Why the authors decide to freeze the previous layers when they add a new layer? The NN that have been used are relatively small.
- I wouldn't call the improvement of electron densities as "remarkable" (page 9). Figure 9 shows that the model fails to provide reasonable densities for specific cases (eg ethanol, water).
- Overall, a model that claims that can represent the density based on electronic structure descriptors and "learn" it from CCSD(T) data should be transferable between atoms with the same size and the same number and type of atomic orbitals (same angular momenta). The authors generated higher expectations in their introduction and fail to present that novelty in their results sections.

Reviewer #2 (Remarks to the Author):

Review comments for authors

This paper represents a novel approach for creating density functionals using machine learning techniques, and so should be published, possibly in Nature Communications. The paper shows promising results in terms of transferability and accuracy for several very different systems. But the presentation needs considerable

revision before a final decision on importance can be made.

- Lots of essential information on the ML part of this paper are missing. The training procedure is totally unclear. For example, what is the loss function being minimized? Is it some combination of density and energy? Or "cross validation" has determined rho to lower generalization errors – what procedure exactly? The authors must carefully describe all these details, in supplemental info if not in the main text. Even the values of r_0 in Eq(1) do not seem to have been reported for each of the functionals used.
- Almost no information is given about computational cost of the method (not cost of data) and how it scales with number of training data.
- The reference to github for supplemental data is insufficient, there should be a DOI for the codes, etc. (see nature guidelines), guaranteeing no changes occur.
- Figure 2 should have been plotted with the usual log scale on the y-axis, without which it is not very useful.
- Figure 3 is hard to comprehend. According to the caption, the target corrections range over about $\pm 0.06\text{eV}$, which is about the same size as the required accuracy (43 meV). So inserting NO correction at all would almost pass this test? This implies the PBE results alone are almost at chemical accuracy for these molecules? What am I missing? (first row of tab 1 seems to say PBE errors are about 35 meV...)
- Fig 4 seems to have been plotted with very few r/r_0 values as one can see straightline segments. There should be more plots, plus supp info listing the numbers and plotting of errors relative to $s66x8$.
- A recent JCP by Paesani and co-workers discusses the decomposition of errors of different functionals. The errors should also be compared with the errors of better functionals than PBE (see Chem. Sci., 2019,10, 8211-8218)
- In Fig 5, the authors even state that their results should not be compared directly to experiment. So they should be compared to MB-pol (without quantum dynamics) to show NeuralXC having much smaller errors than PBE. Again, comparison with a better functional for water should also be included.
- In Fig 6, why is NeuralXC compared to PBE, and not to CCSD(T)? It is almost impossible to judge, with the present pictures, if NeuralXC is doing a significantly better job for the density.
- In several places, the need for methods to work in the thermodynamic limit is mentioned, but none of the examples seem to show this. Either this should be demonstrated or removed.

Reviewer #3 (Remarks to the Author):

The paper presents a further step forward in the development of machine-learned exchange-correlation functionals. The authors build on previously successful ideas such as descriptors based on a basis decomposition of the density and Behler-type neural networks to produce NN-based XC functionals which in certain cases achieve beyond chemical accuracy. Self-consistent calculations are performed with the fitted functionals, which is a strong point of the paper. ML functionals are now actively researched, successes in this field and eventual deployment of ML functionals in end-user codes would be a big step forward for computational chemistry and computational materials science. For this reason, the paper belongs in the literature, and I would recommend eventual acceptance after a revision that needs to address the following:

- The sentence "In this manuscript, we propose a pathway to construct fully machine-learned functionals that depend explicitly on the electronic density." could be misleading, as what is done is actually delta-learning and the functional does not depend only on the density but also on atomic positions. This should be highlighted early on as this provides a ML algorithm major help which can be decisive e.g. it is atomic position information that allowed previous NN works, albeit on KEF, to get smooth potential curves (CPL 734 (2019) 136732).

- Details of NNs should be given explicitly. How do the results, and especially transferability, depend on the details of atomic NN architectures? (nos. of layers and neurons and their type). It was previously reported that NN architecture can critically influence transferability (PCCP 21 (2019) 378), so this is potentially an important issue and should be looked into.

- Was SIESTA DZP basis default or tuned? Default basis functions in SIESTA can be very bad, I wonder if that could color the results.

Reviewers' comments:

Reviewer #1 (Remarks to the Author):

Dick and Fernandez-Serra discuss the development of density functionals based on supervised machine learning by using CCSD(T) energies as reference. Their aim is to provide a highly-accurate density (and energy) at the speed of a calculation performed with a GGA functional. In this manuscript, they present an interesting method that uses atomic orbitals as feature space of the input electron density. A neural network is then trained that can predict a new, more accurate density. However, the method has already been introduced in a previous publication, while the two manuscripts share many similarities, including test examples and figures. In addition, the authors mention in every section (including the title) the word “accuracy”, but their results do not fully reflect that statement. For those reasons, I cannot recommend this article for publication in Nature Communications.

Technical comments:

We thank the referee for his comments and careful evaluation of our work. We understand that he/she does not recommend the work for publication, however we would still like to address the comments in the hope that the referee might reconsider the current assessment of the manuscript:

- The title is misleading. Reference data from CCSD(T) obtained with a triple-zeta quality basis set should not be called as “accurate”. There might be more accurate than GGA functionals, but in order to obtain the expected accuracy of post-HF methods and especially coupled-cluster methods, a complete basis set is needed (basis set extrapolation, explicitly correlated methods).

We disagree with this interpretation.

First we would like to point out that in this context, “accurate” should be interpreted as “accurate with respect to the reference data”. Our use of the term “accurate” is in line with other publications on machine learned force fields (see e.g. Chmiela et al., Nat Commun. 2018 Sep 24;9(1):3887).

Of course, the final functional will be limited by the accuracy of the target data, however, this accuracy can in principle be set to be as high as needed at the time of the data generation. Moreover studies by Cheng et al. (L. Cheng, M. Welborn, A. S. Christensen, and T. F. Miller III, JCP 150, 131103 (2019)) suggest that the level of reference method used (e.g. MP2 vs. CCSD(T)) does not have any significant influence on the fitting accuracy. Chmiela et al. (Nat Commun. 2018 Sep 24;9(1):3887) even observe that higher-level reference calculations (CCSD(T)) are easier to fit, as their potential energy surfaces seem to be smoother than those of lower-level methods (DFT).

If we had used the word “exact” in the title, then we would agree that it would not be a fair representation. However, the meaning of accurate does apply to all the functionals and data that we present in this work.

- I do not understand why the authors are trying to map a GGA density generated with planewaves and pseudopotentials to data from cluster CCSD(T) calculations. Since all calculations have been performed for molecular species, why they do not use a cluster DFT code?

We have chosen to implement NeuralXC in SIESTA, as our main goal is to be able to use the machine-learned functionals to study condensed systems. Due to the fact that SIESTA uses strictly localized atomic orbitals as basis sets (it is not a plane wave code), it is very versatile in both the treatment of clusters and periodic systems. In spite of using periodic boundary conditions the unit cell can in principle be chosen as large as needed for cluster calculations, without any significant added computational cost. The use of pseudopotentials is standard in the treatment of large, condensed systems, but our method itself - being a density functional - is independent of the presence of pseudopotentials and the kind of basis used.

We think that some of the referee’s concern is a result of our lack of explanation in the manuscript. Hence, we have added the following sentences to serve as clarification on this point.

“Siesta is a DFT code that uses strictly localized atomic orbitals as basis set. This allows us to easily perform cluster calculations, because, despite it being a periodic code, we can make our unit cells as large as needed. This choice was made both because of its implementation advantages and because one of the main applications of this method is to treat accurately condensed systems.”

All that being said, we are currently working on a version of the method adapted to PySCF (a cluster code). However, we do not think this is a requirement to have the current method published.

- The computational details for the CCSD(T) calculations are pretty messy. No quantum chemical code is mentioned for the generation of reference data for the sGCML and MB-pol. In addition, using different reference data from different codes and approximations (eg. density fitting), makes the analysis of the results imbalanced.

We have added information about the quantum chemical codes used at the appropriate section. Having done so, we believe that enough information is contained in the “Datasets” section for the reader to get a general understanding of the methods used to create the

reference data. For more details, we refer the reader to the appropriate sources, as every dataset we used is already well documented in their original publication.

Regarding the second comment, we believe that there is no imbalance, given that for all the different NeuralXC-functionals that we present in this work, training and test sets were always obtained with the same reference method. Only in the case of water, we make transferability tests with two different data sets (MB-pol and s66x8). However in this case, both sets have been obtained at the CCSD(T) level at the complete basis set limit using MOLPRO.

- Efforts to satisfy invariance by data augmentation is inefficient.

We agree with the referee on this point which is why we have, in fact, not used data augmentation to satisfy invariance. However, upon re-reading our manuscript, we understand how the original phrasing might have led to some confusion. We have therefore re-written the section for clarification.

- In the section “data-efficiency”, I don’t understand where the efficiency is if you need a dataset of at least 500 samples to reach 1 kcal/mol (malonaldehyde).

The goal of this section is not to claim the excellent data efficiency of our method. Instead we simply present the efficiency that the method achieves. While in general the method seems to be rather efficient, there is still room for improvement, as in the case of malonaldehyde.

- Abstract and introduction mention that density functionals have been developed, so I was expecting one unified functional as the outcome of the ML methodology. On the contrary, the manuscript describes the development of case-specific functionals.

The referee raises a valid and important point. This work is motivated by the view expressed in reference [M. Fritz, M. Fernandez-Serra, and J. M. Soler, *The Journal of chemical physics* 144, 224101 (2016)], where the goal of optimizing a density functional for a specific system is prioritized over its transferability. In our opinion, this perspective is a required starting point to improve the construction of new density functionals using machine learning methods. We believe that our results regarding transferability show potential for achieving universality. However, this is a direction that requires an effort beyond the scope of this manuscript and we therefore aim to pursue it in future work. In order to clarify this, we have added a sentence in the last paragraph of the introduction section.

- The example of cats and dogs should be removed.

We have removed the example.

- Why the authors decide to freeze the previous layers when they add a new layer? The NN that have been used are relatively small.

Freezing the previous layers has proven beneficial for the convergence of the iterative training procedure. We have added a discussion of this in the Supplementary Information under “Model Training and Hyperparameters”.

- I wouldn't call the improvement of electron densities as "remarkable" (page 9). Figure 9 shows that the model fails to provide reasonable densities for specific cases (eg ethanol, water).

We agree with the referee that there is room for improvement in this point. Again, we believe that this is work to be done in the future .

We have changed the sentence to:

"The improvement of electron densities in some cases is remarkable"

We have also modified the original sentence in the conclusion section to:

"This work has put an emphasis on energetic properties, but we have shown that by correcting the total baseline energy NeuralXC also induces density changes that bring the density closer to the exact density. These changes are however relatively small and only present in a subset of tested systems"

- Overall, a model that claims that can represent the density based on electronic structure descriptors and "learn" it from CCSD(T) data should be transferable between atoms with the same size and the same number and type of atomic orbitals (same angular momenta). The authors generated higher expectations in their introduction and fail to present that novelty in their results sections.

We do not understand this comment, given that different atomic species have different sizes. We have shown transferability for the same species in different molecules. However, transferability between different atoms is not shown, because at this point the functional is atom specific. This is something that will be part of the work we aim to do to achieve universality.

We have tried to address the referee's concerns to the best of our abilities. We have also largely improved the manuscript with responses to this and other referee's comments. We hope that the referee would be willing to re-evaluate the work after these improvements.

Reviewer #2 (Remarks to the Author):

Review comments for authors

This paper represents a novel approach for creating density functionals using machine learning techniques, and so should be published, possibly in Nature Communications. The paper shows promising results in terms of transferability and accuracy for several very different systems. But the presentation needs considerable revision before a final decision on importance can be made.

We thank the referee for the positive comments, requests and suggestions. We agree that modifications were needed to both the presentation and content of the manuscript. We have tried to address all the raised points to the best of our ability as follows:

- Lots of essential information on the ML part of this paper are missing. The training procedure is totally unclear. For example, what is the loss function being minimized? Is it some combination of density and energy? Or “cross validation” has determined ρ to lower generalization errors – what procedure exactly? The authors must carefully describe all these details, in supplemental info if not in the main text. Even the values of r_0 in Eq(1) do not seem to have been reported for each of the functionals used.

We have added information to the Methods section to address these concerns. We have also dedicated a section in the supplemental information to model training and hyperparameter selection.

The cutoff radius r_0 as well as other basis set parameters appearing in Eq(1) were already provided in a table that has been moved to the SI in the revised version of this manuscript (Table I). The table was moved after we added substantial information regarding the network architecture following reviewer three’s request.

- Almost no information is given about computational cost of the method (not cost of data) and how it scales with number of training data.

We would like to point out that the cost of evaluating a trained neural network is independent of its training set size (as opposed to kernel-based methods). We have added this information along with a plot of the computational cost of the method with respect to system size in the SI (Fig. 3).

- The reference to github for supplemental data is insufficient, there should be a DOI for the codes, etc. (see nature guidelines), guaranteeing no changes occur.

Following Nature guidelines we have now created a DOI for both the NeuralXC implementation and the reference/supplemental data (<http://doi.org/10.5281/zenodo.3637602>)

- Figure 2 should have been plotted with the usual log scale on the y-axis, without which it is not very useful.

We have changed the axes in Figure 2 to use a log scale.

- Figure 3 is hard to comprehend. According to the caption, the target corrections range over about ± 0.06 eV, which is about the same size as the required accuracy (43 meV). So inserting NO correction at all would almost pass this test? This implies the PBE results alone are almost at chemical accuracy for these molecules? What am I missing? (first row of tab 1 seems to say PBE errors are about 35 meV...)

We agree that the caption was not clear enough, we have therefore largely modified it. The confusion regarding the target correction range might also be due to the fact that we use different scales: subplot a ranges from -0.15 meV to 0.10 meV and subplot b ranges from -0.06 meV to 0.06 meV.

We agree that the systems depicted in subplot b) almost have the required accuracy if calculated with PBE. We opted to include these systems, not necessarily to show the accuracy of our method but to illustrate challenges that arise when the model is used on other carbon hybridizations (systematic errors that show in the linear correlation in subplot b as discussed in the manuscript).

- Fig 4 seems to have been plotted with very few r/r_0 values as one can see straightline segments. There should be more plots, plus supp info listing the numbers and plotting of errors relative to s66x8.

In the process of addressing this comment we realized that the data for PBE presented in Fig 4 was incorrectly labeled. Instead the data shown in the original figure was for vdW-cx. We have recalculated all the systems with PBE and the results show that PBE is very close to CCSD(T) in these H-bonded systems. Because of this, while the NXC results (which remain unchanged) still improve over PBE results on average, the margin of improvement is very much reduced. We hence have opted for moving this figure and its discussion to the SI. Doing so we have extended the discussion to the entire s66 dataset instead of only using a subset as in the original submission of the manuscript. The formatting of the binding plot curves was modified for clarity.

- A recent JCP by Paesani and co-workers discusses the decomposition of errors of different functionals. The errors should also be compared with the errors of better functionals than PBE (see Chem. Sci., 2019,10, 8211-8218)

In addition to adding a reference to the paper mentioned, we have included a comparison with SCAN and ω B97M-V, two state of the art functionals.

- In Fig 5, the authors even state that their results should not be compared directly to experiment. So they should be compared to MB-pol (without quantum dynamics) to show NeuralXC having much smaller errors than PBE. Again, comparison with a better functional for water should also be included.

We agree with the referee. Nonetheless it is standard practice to include experimental results in radial distribution function plots of water, for reference. Hence we have left the experimental data in the plot. In addition we have also included a comparison with MB-pol and two additional

density functionals (SCAN and ω B97M-V) in the SI. These are chosen from the literature, from works that use room temperature and classical nuclear dynamics.

- In Fig 6, why is NeuralXC compared to PBE, and not to CCSD(T)? It is almost impossible to judge, with the present pictures, if NeuralXC is doing a significantly better job for the density.

As explained in the manuscript, our NXC implementation is done within Siesta, a code that uses pseudo-potentials and different basis sets than Pyscf, our reference code. Hence we choose to compare density differences, with PBE as reference. This removes the effects of basis sets and pseudopotential which would mask the quality of NXC results.

While this was already stated in the manuscript we have rewritten it to make it more clear.

- In several places, the need for methods to work in the thermodynamic limit is mentioned, but none of the examples seem to show this. Either this should be demonstrated or removed.

We presume that the referee is indicating that we claim to create a functional that can be used in larger (condensed) systems, despite being generated using data for gas phase systems (molecules and clusters). We believe that the demonstration for this is in the example of liquid water. We show that a functional can be used in a condensed system, despite having been optimized only in very small (up to 3 water molecules only) clusters.

We have now indicated this in the manuscript. The molecular dynamics section is renamed Condensed systems and Molecular Dynamics, and an explicit description of this size extensivity is included.

We thank the referee for helping to improve the quality of the work and hope that the current version of the manuscript could be constructively reevaluated for a final decision.

Reviewer #3 (Remarks to the Author):

The paper presents a further step forward in the development of machine-learned exchange-correlation functionals. The authors build on previously successful ideas such as descriptors based on a basis decomposition of the density and Behler-type neural networks to produce NN-based XC functionals which in certain cases achieve beyond chemical accuracy. Self-consistent calculations are performed with the fitted functionals, which is a strong point of the paper. ML functionals are now actively researched, successes in this field and eventual deployment of ML functionals in end-user codes would be a big step forward for computational chemistry and computational materials science. For this reason, the paper belongs in the literature, and I would recommend eventual acceptance after a revision that needs to address the following:

- The sentence "In this manuscript, we propose a pathway to construct fully machine-learned functionals that depend explicitly on the electronic density." could be misleading, as what is done is actually delta-learning and the functional does not depend only on the density but also on atomic positions. This should be highlighted early on as this provides a ML algorithm major help which can be decisive e.g. it is atomic position information that allowed previous NN works, albeit on KEF, to get smooth potential curves (CPL 734 (2019) 136732).

The referee raises a very important point, and we have indicated this early on in the manuscript:

“In this manuscript, we propose a pathway to construct fully machine-learned functionals that depend explicitly on the electronic density and implicitly on the atomic positions and are built on top of physically motivated functionals in a delta-learning type approach.”

We have also included an explicit reference to (CPL 734 (2019) 136732) in the section where we describe the construction of the exchange and correlation functional.

- Details of NNs should be given explicitly. How do the results, and especially transferability, depend on the details of atomic NN architectures? (nos. of layers and neurons and their type). It was previously reported that NN architecture can critically influence transferability (PCCP 21 (2019) 378), so this is potentially an important issue and should be looked into.

We have now added a new section in the Supplementary information that includes calculations that explicitly discuss this point. In particular, we have now evaluated the transferability of the method as a function of its architecture for small alkanes (MOB-ML dataset).

- Was SIESTA DZP basis default or tuned? Default basis functions in SIESTA can be very bad, I wonder if that could color the results.

We agree that a lot of care needs to be taken with SIESTA basis sets. All our calculations are done with well converged basis sets. However, the referee is correct on pointing out this issue. Being a Delta functional, NeuralXC is designed to correct for errors that, in the complete basis set limit should mostly originate from the XC approximation. However, any DFT implementation has other numerical errors, including, but not limited to those arising from grids, pseudopotentials, and convergence tolerances. We, however, demonstrate in our work that a functional that has been optimized using a set of fully converged parameters (quadruple-Z polarized basis for liquid water) can be used with a less accurate basis (double-Z polarized) and achieve the required XC correction. Hence our results indicate that the fitting of the functional should be done with a highly converged set of parameters, but the corresponding functional can then be transferable in the numerical parameter space.

We have also now provided in the Supplementary material all the SIESTA input files we have used in the different sections, so as calculations can be reproduced if necessary.

We hope the referee will be satisfied with all the modifications we have done to the manuscript. The new S.I. section contains abundant information that we agree was necessary to make the work more understandable and reproducible.

Reviewers' comments:

Reviewer #1 (Remarks to the Author):

The authors have responded to the comments raised by all reviewers. Most of their responses are satisfactory. However, there are still a few points that are not clear or they need additional attention by the authors:

- I disagree with the argument that "different atomic species have different (basis) sizes". For a given basis (eg. cc-pVTZ), the number of basis functions is constant for atoms that are in the same row of the periodic table. For example, cc-pVTZ has 30 basis functions for atoms Li-Ne ([https://en.wikipedia.org/wiki/Basis_set_\(chemistry\)#Correlation-consistent_basis_sets](https://en.wikipedia.org/wiki/Basis_set_(chemistry)#Correlation-consistent_basis_sets)). Based on that, and since NeuralXC is using atom-based functions, I was expecting transferability between molecules that have different atoms of the same row.

- I am still concerned about the applicability of NeuralXC. The authors now show in Figure 4 and Table II of the SI that PBE and NeuralXC have the same errors for the S66x8 dataset. What NeuralXC learns at the end?

Reviewer #2 (Remarks to the Author):

The authors have responded to all the points raised by the referees, and the manuscript is much improved. However, some of the improvements in presentation have lessened the possible significance of the work. And further improvements are still required (see below). But once these are done, and if all questions have satisfactory answers, it should be acceptable for Nat Comm

1. The results on the S66 test set are entirely unconvincing and should be eliminated or mentioned only as "no significant improvement over PBE was found." If the authors compared them with any state of the art functional, it would be clear that their numbers for NeuralXC are irrelevant.
2. The results for densities are also less than convincing, since they consist of colored patches. The density differences should be quantified, to show that the NeuralXC densities are significantly closer to the CCSD(T) densities than either are to PBE. But the real problem is that in two cases, ethanol and water, they are not. In the absence of explanation of why not in those cases, this cannot be used to claim efficacy of the method, and all such claims should be removed (or bolstered). It is particularly disturbing when the primary result of significance is the improved energetics for water clusters and in condensed phase.
3. In the hydrocarbon tests, it would be useful to consider error per bond instead of total error, in case it is just a question of system size. Also, it should be easy to include one or two double/triple bonds in the training set, and then see if that immediately reduces the linear component of the error in 3(b), justifying the authors' hypothesis
4. Much of the abstract, introduction, and conclusions is spent on very general statements, but nowhere do the authors give a precise description of what useful calculations their methodology should now make possible. Extrapolating the accuracies shown here, could I now, e.g., calculate the potential of mean force for methane or ethanol in water, at PBE cost but with CCSD(T)-like accuracies, using NeuralXC? With so many qualifiers on so many of their statements, or lack of precision, it is not possible for a reader to tell. Moreover, given that MB-pol works for water, and the costs are surely greater than running MB-pol, pointing out applications beyond water is crucial.
5. The authors insist on showing only the comparison with PBE, experiment, and their method in the main text for the water structure factors, but the convincing comparison is with the MB-POL results, and showing that NeuralXC does much better than all the other functionals shown. This should be in the main text. Some simple integrated measure of the errors should be reported, to see if there are any measurable differences with MB-POL. This again begs the question of what NeuralXC provides that MB-POL does not?

We thank both referees for their additional questions and suggestions. We believe that the changes and additions we included after this round have very positively improved the manuscript. We hope these changes will satisfy all the referee's objections to publishing the manuscript. All the changes/additions in the manuscript appear in blue. New figures and tables are pointed to in the different responses. We have added Section IV and VI to the SI and overhauled Section V - S66 dataset

Reviewers' comments:

Reviewer #1 (Remarks to the Author):

The authors have responded to the comments raised by all reviewers. Most of their responses are satisfactory. However, there are still a few points that are not clear or they need additional attention by the authors:

- I disagree with the argument that "different atomic species have different (basis) sizes". For a given basis (eg. cc-pVTZ), the number of basis functions is constant for atoms that are in the same row of the periodic table. For example, cc-pVTZ has 30 basis functions for atoms Li-Ne ([https://en.wikipedia.org/wiki/Basis_set_\(chemistry\)#Correlation-consistent_basis_sets](https://en.wikipedia.org/wiki/Basis_set_(chemistry)#Correlation-consistent_basis_sets)). Based on that, and since NeuralXC is using atom-based functions, I was expecting transferability between molecules that have different atoms of the same row.

We realize that our last response to this concern was based on a misunderstanding on our side and we thank the referee for providing more details to clear this up .

We agree that the size of a basis set is, in general, the same across elements in the same row of the periodic table.

However, we believe that the key to a transferable model does not necessarily lie in the basis sets used but in the diversity of chemistries included in the training data. For example, we expect better transferability between C and Si than between C and O. This is because equal valence atoms will tend to have similar lewis pair bond structures and hence similar density distributions. It is this density distribution that our model draws upon to make inferences on the system's exchange and correlation energy.

In order to confirm our hypothesis, we have created a new species-independent model, trained with C and O-based molecules that was then tested on Si and S-based molecules. In spite of using a relatively crude training set (it is simply an agglomeration of all the molecules previously used in the paper) the results are encouraging. We have added a summary in the main manuscript and extended the SI by a new section: IV Intragroup Transferability.

- I am still concerned about the applicability of NeuralXC. The authors now show in Figure 4 and Table II of the SI that PBE and NeuralXC have the same errors for the S66x8 dataset. What NeuralXC learns at the end?

The Figure and Table that the referee is referring to, show the RSM error in the binding energy with respect to CCSD(T) results. These results were obtained averaging over 8 different intermolecular pair distances for all the different pairs of molecules shown in the table.

The model used for this - our water model NXC-W01 - is outstanding in the treatment of water, as shown in the main manuscript (including the new results shown in Fig. 5). It learns an XC potential capable of producing CCSD(T)-level results both at the energetic and structural level.

When used on the S66 set, a dataset which contains, but is not limited to water, the above results indicate that the model does not degrade with respect to its baseline, PBE. This in itself is a positive result: It ensures that the model can be used to describe heterogeneous systems, where significant improvement is obtained in the description of water-water interactions, with no decrease of accuracy for the other molecules. When it comes to water this is very important, given that treating water as a solvent is the challenging part within a full DFT description.

Furthermore, we believe that the metric initially chosen by us (RMS error over 8 different distances) might not have been an ideal proxy to reflect the improvement of H-bond distances, as energies closer to the equilibrium distance carry more weight in determining the value thereof.

To account for this and both the referees' comments, we have replaced Table II by what is now Table III in the SI. This new table contains equilibrium H-bond distances rather than RMS errors in energy. The improvement of the error from PBE (2% average error) to NeuralXC (1% average error) is significant. The original energy errors (averaged over the dataset) are still mentioned in the text.

These results demonstrate several things about the model:

- (i) The model learns an XC functional that is highly accurate for liquid water. (See results for water in the main manuscript)*
- (ii) The model is transferable to systems/situations which it has not been trained on. (See OH bond breaking in Fig 5 of main paper and improved H-bond geometries for the S66 data set in SI Table III)*
- (iii) In cases where NeuralXC does not improve results (e.g. dispersion bonded geometries of the s66 dataset), our results indicate that it does not degrade with respect to its baseline.*

Reviewer #2 (Remarks to the Author):

The authors have responded to all the points raised by the referees, and the manuscript is much improved. However, some of the improvements in presentation have lessened the possible significance of the work. And further improvements are still required (see below). But once these are done, and if all questions have satisfactory answers, it should be acceptable for Nat Comm

1. The results on the S66 test set are entirely unconvincing and should be eliminated or mentioned only as "no significant improvement over PBE was found." If the authors compared them with any state of the art functional, it would be clear that their numbers for NeuralXC are irrelevant.

We have made changes and provided an extended answer to this point in referee one's comment #2. We refer to this answer right above.

2. The results for densities are also less than convincing, since they consist of colored patches. The density differences should be quantified, to show that the NeuralXC densities are significantly closer to the CCSD(T) densities than either are to PBE. But the real problem is that in two cases, ethanol and water, they are not. In the absence of explanation of why not in those cases, this cannot be used to claim efficacy of the method, and all such claims should be removed (or bolstered). It is particularly disturbing when the primary result of significance is the improved energetics for water clusters and in condensed phase.

The referee is right on pointing to this important issue. While it is not necessarily true that an improved density functional always improves on the density [<https://science.sciencemag.org/content/355/6320/49>], we initially wanted to investigate whether this is the case with NeuralXC. In particular, we were aiming to understand if using density-based descriptors in a self consistent method would automatically encode some information about the exact density, even if this was not explicitly included in the training process. Our initial results indicate that (i) improvement is seen in specific cases only (ii) improvement is achieved when a functional is trained with large amounts of highly accurate and carefully curated data. In our case this means that the water model (NXC-W01) presents measurable improvement with respect to the baseline (PBE). We have made the following addition/changes to the manuscript:

(i) We have revisited the change in electron density for the water model (NXC-W01) by re-calculating both the DFT density as well as the coupled cluster density with a larger basis set (quadruple zeta in the case of DFT and aug-cc-pVTZ for coupled cluster). The updated results are presented in the main text. While there are still undeniable differences in the electron density (Fig. 6), especially around the oxygen core, the moments of the density (dipole, quadrupole moment) show good agreement with both experiment and reference calculation. We have added an explanation for the deviations in density in lines 624-642.

(II) We have changed the introduction, lowering the expectations of the method on reproducing the exact density [see lines 115-123].

(III) Understanding the relevance of this point we have explored a pathway to incorporate in our method a way to enforce the reproduction of the exact density, in addition to the exact total energy. We propose to upgrade the loss function of the training to include the exact potential (the effective one body potential that can be calculated once the exact density is known). This method is presented in the SI (VI: Fitting to the exact potential). We however think that this new section represents a complete new work on its own, and hence we only provide a proof of concept example.

We hope these modifications and important additions are enough to satisfy the referee's objections in this point. We particularly thank the referee, because we believe that these changes have significantly improved the quality of the manuscript.

3. In the hydrocarbon tests, it would be useful to consider error per bond instead of total error, in case it is just a question of system size. Also, it should be easy to include one or two double/triple bonds in the training set, and then see if that immediately reduces the linear component of the error in 3(b), justifying the authors' hypothesis

We agree with the referee on this point. As our functional correction is computed as a sum of local corrections, it is instructive to consider the errors per carbon atom. We have adjusted Table I accordingly. In addition to providing the error per c-atom we have included data for ethane and propane test sets; unfortunately results for MOB-ML for these systems were not available.

Following the reviewer's suggestion we have been able to confirm our hypotheses that including a small sample of double/triple bonds reduces the linear component of the error in 3(b). We have added a discussion in the main text (line 426-436)

4. Much of the abstract, introduction, and conclusions is spent on very general statements, but nowhere do the authors give a precise description of what useful calculations their methodology should now make possible. Extrapolating the accuracies shown here, could I now, e.g., calculate the potential of mean force for methane or ethanol in water, at PBE cost but with CCSD(T)-like accuracies, using NeuralXC? With so many qualifiers on so many of their statements, or lack of precision, it is not possible for a reader to tell. Moreover, given that MB-pol works for water, and the costs are surely greater than running MB-pol, pointing out applications beyond water is crucial.

Addressing this valid concern we have toned down statements in the abstract and discussion by replacing the phrase "lift the accuracy [...] to that provided by more accurate methods" by "lift the accuracy [...] toward that ...", as reliably, we can only observe coupled-cluster-like accuracies for systems similar to those contained in the training set.

We have further added sections in the Introduction (ll. 103-111) and the Discussion (ll. 667-673) outlining the capabilities (and limits) of NeuralXC in more detail.

We have also explored this issue with additional calculations. In particular, driven by the question of what we can do beyond MB-Pol we have evaluated how NXC-W01 performs in a number of situations which are out of the scope of MB-Pol. While believe that parts of these concerns are already answered in referee one's point #2, we would like to further highlight the following points:

(l) H-bond interactions of water with other molecules: Table III in the SI shows results for equilibrium geometries obtained with NXC-W01 compared to PBE and to MP2 (most accurate results available for these geometries). Overall there is a 50% improvement of the error from

PBE to NXC-W01. While PBE is already fairly accurate, what these results show is that NXC-W01 can accurately treat water-based systems containing other organic molecules. Indeed, having a functional that works well for water, while not degrading in the treatment of other components is useful. Problems that involve understanding the structure and dynamics of ions in liquid water water-metal interactions or water-semiconducting interfaces can already be addressed with this functional. Perhaps the next point addresses this in more detail.

(II) We have evaluated the capacity of NXC-W01 to reproduce bond breaking situations. New Figure (Fig.5) in the manuscript shows the energy barrier for the coordinated proton transfer reaction in a water Hexamer ring. We chose this system because it is one where DFT struggles with, as shown by the results presented for other functionals (including Hybrid functionals). The model was not trained for any bond breaking situation, and yet it very closely reproduces CCSD(T) results. This might seem surprising, as machine learned models generally do not extrapolate beyond their training set. However, what this results indicates is that, by using a semi-local density representation as descriptors, it is possible to extrapolate in configurational space, while still remaining within the bounds of valid input/feature space. As indicated in the text (lines 564-580) this would make other simulations like path integral molecular dynamics with this model more accurate than with other methods.

5. The authors insist on showing only the comparison with PBE, experiment, and their method in the main text for the water structure factors, but the convincing comparison is with the MB-POL results, and showing that NeuralXC does much better than all the other functionals shown. This should be in the main text. Some simple integrated measure of the errors should be reported, to see if there are any measurable differences with MB-POL. This again begs the question of what NeuralXC provides that MB-POL does not?

We have moved the comparison of radial distribution functions to those obtained with MB-POL from the SI to the main text. We feel that the integrated measure of error would not be sufficient. There are cutoffs involved that are still method dependent (the coordination number involves the integral up to the first minimum of the RDF), and the numbers obtained this way are subject to errors. Because of this it is not a standard practice in the literature to include a comparable metric in the study of radial distribution functions and we are unsure what would be gained by computing such value. We believe that a visual comparison of the rdfs provides the most insight. Small deviations between MB-Pol and NXC-W01 can be observed in the OO RDF function between 3.5 and 4.5 angstroms.

For the last question we refer to our answer to point (4).

REVIEWERS' COMMENTS:

Reviewer #1 (Remarks to the Author):

The authors provided an updated manuscript where they have responded to all comments, most of their bold statements about their method have been rewritten in order to reflect the true capabilities, while some of the issues that have been identified by the reviewers were not addressed but left as future work. I am also confused about Table II. They mention that they "have replaced Table II by what is now Table III in the SI", but table II is still in the main document. I do not have any further comments.

Reviewer #2 (Remarks to the Author):

The authors have done a reasonably good job of addressing the issues raised by the referees, and with the additional examples generated, this paper should now be accepted for publication.

Two last points should be considered:

1. The abstract was not revised as requested. It still lacks specific detail about what has been accomplished in this work, and what a reader may hope to use the method for. 80% of it is background generalities.
2. There is quite extensive new material and discussion, some of it confined to the supp info and briefly mentioned in the main text. These mentionings should be more specific (say where in supp info the material is) and describe the results and conclusions in a sentence or two in the main text.

I leave these suggestions to the authors and editor.

We thank the referees for their final comments. Here we provide a summary of all the modifications made to the final version of the manuscript before publication.

The manuscript has been modified following the editorial guidelines of the journal. In particular, the abstract of the manuscript has been shortened to less than 150 words and the full text has undergone a substantial shortening. All of the removed sections now appear within the Supplementary Information (details provided in a separate document). In addition, we have provided a new version of Fig. 5, with a new radial distribution function for our NXC model. The new results do not modify our original conclusions, but they are obtained for a longer molecular dynamics trajectory computed with a more reliable thermostat. We have added the details for the new simulation in the text.

Reviewer #1 (Remarks to the Author):

The authors provided an updated manuscript where they have responded to all comments, most of their bold statements about their method have been rewritten in order to reflect the true capabilities, while some of the issues that have been identified by the reviewers were not addressed but left as future work. I am also confused about Table II. They mention that they "have replaced Table II by what is now Table III in the SI", but table II is still in the main document. I do not have any further comments.

Table II in the main document was not modified. All the changes previously cited made reference to the SI. We added a new table, Supplementary Table II, and what used to be Supplementary Table II modified and is now listed as Supplementary Table III. We apologize for the confusion.

Reviewer #2 (Remarks to the Author):

The authors have done a reasonably good job of addressing the issues raised by the referees, and with the additional examples generated, this paper should now be accepted for publication.

Two last points should be considered:

1. The abstract was not revised as requested. It still lacks specific detail about what has been accomplished in this work, and what a reader may hope to use the method for. 80% of it is background generalities.

In shortening the abstract to less than 150 words we have now addressed the concerns originally pointed by the referee.

2. There is quite extensive new material and discussion, some of it confined to the supp info and briefly mentioned in the main text. These mentionings should be more specific (say where in supp info the material is) and describe the results and conclusions in a sentence or two in the main text.

We have now clarified where in the SI the material can be found according to Nat. Commun. Guidelines. Regarding the results concerning transferability we have added the following sentences:

“In particular, the average error in bond lengths for a set of small molecules decreased by approximately 42%“, regarding Supplementary Note 5 and

“While we aim to explore this in more detail in future work, we provide a proof of concept example in Supplementary Note 7, showing how the density error of PBE can be decreased by up to two orders of magnitude for a set of H₂ molecules.” , regarding Supplementary Note 7.